# Changes in Brain Activation through Cognitive-Behavioral Therapy with Exposure to Virtual Reality: A Neuroimaging Study of Specific Phobia

**DOI:** 10.3390/jcm10163505

**Published:** 2021-08-09

**Authors:** Yolanda Álvarez-Pérez, Francisco Rivero, Manuel Herrero, Conrado Viña, Ascensión Fumero, Moisés Betancort, Wenceslao Peñate

**Affiliations:** 1Canary Islands Health Research Institute Foundation (FIISC), 38109 Tenerife, Spain; 2Department of Clinical Psychology, Psychobiology and Methodology, University of La Laguna, 38200 Tenerife, Spain; friverop@ull.edu.es (F.R.); mherrero@ull.edu.es (M.H.); cmvinalo@ull.edu.es (C.V.); afumero@ull.edu.es (A.F.); moibemo@ull.edu.es (M.B.); wpenate@ull.es (W.P.); 3University Institute of Neurosciences (IUNE), University of La Laguna, 38200 Tenerife, Spain

**Keywords:** specific phobia, virtual reality, cognitive-behavioral therapy, magnetic resonance imaging

## Abstract

Background: Cognitive-behavioral therapy (CBT) with exposure is the treatment of choice for specific phobia. Virtual reality exposure therapy (VRET) has shown benefits for the treatment and prevention of the return of fear in specific phobias by addressing the therapeutic limitations of exposure to real images. Method: Thirty-one participants with specific phobias to small animals were included: 14 were treated with CBT + VRET (intervention group), and 17 were treated with CBT + exposure to real images (active control group). Participants’ scores in anxiety and phobia levels were measured at baseline, post-treatment, and 3-month follow-up, and brain activation was measured through functional magnetic resonance imaging (fMRI) baseline and post-treatment. Results: Both groups showed a significant decrease in anxiety and phobia scores after the therapy and were maintained until follow-up. There were no significant differences between both groups. Overall, fMRI tests showed a significant decrease in brain activity after treatment in some structures (e.g., prefrontal and frontal cortex) and other structures (e.g., precuneus) showed an increasing activity after therapy. However, structures such as the amygdala remained active in both groups. Conclusions: The efficacy of CBT + VRET was observed in the significant decrease in anxiety responses. However, the results of brain activity observed suggest that there was still a fear response in the brain, despite the significant decrease in subjective anxiety levels.

## 1. Introduction

Phobias are one of the most frequent types of anxiety disorders [1]. The clinical relevance of specific phobia is that it is a disease with a chronic course that reduces the coping skills and quality of life of patients in the presence of the phobic stimulus and in the absence of the stimulus, e.g., by avoidance [2,3]. The prevalence of specific phobia worldwide has been estimated at 7.4% and 4.8% specifically for Spain [4]. It is one of the most common and prevalent anxiety disorders present in middle- and high-income countries [5], being less frequent in adults (3–5%) than in children and adolescents (11%) [1]. Similar to most anxiety disorders, specific phobias are more common in women (9.8%) than in men (4.9%) [1,4,6,7]. Animal-specific phobia is the most prevalent phobia subtype [7]. Seventy-five percent of people with this disorder often have a phobia of more than one object or situation [1,8]. As a result, it is conceivable that an underlying general vulnerability enhances the risk for additional comorbid phobias and other anxiety disorders [9,10,11].

The treatment of choice for specific phobia is cognitive-behavioral therapy (CBT) with exposure [12,13]. The exposure can be in vivo, in imagination, or in images. Although in vivo exposure has been shown to be the most effective, the choice of exposure in images is preferable in those cases where in vivo exposure is difficult to manipulate by the therapist or where the patient shows serious reluctance to be exposed to real phobic stimulus [14]. This represents a therapeutic limitation for in vivo exposure, since the patient must be progressively exposed to those phobic stimuli that precisely generate a lot of distress, causing this procedure to benefit only a limited number of patients and to generate considerable dropouts [14,15]. Due to this substantial gap in treatment, it is necessary to consider alternative means of delivering exposure-based treatments [15]. Exposure to images is an acceptable alternative to in vivo exposure, with clinically significant results in various types of specific phobia [16,17,18].

The development of virtual environments is an important improvement considering the therapeutic limitations of exposure to images because it provides three-dimensionality [19,20,21]. Different research supports the clinical efficacy of virtual reality in mental health for conditions including anxiety disorders and others [22,23]. This has led to a treatment known as virtual reality exposure therapy (VRET) [24]. VRET has shown better performance when conducted in a variety of settings and contexts to facilitate generalization and prevent the return of fear in phobia [17,25], and the design goal of achieving virtual reality experiences for treatment in mental health requires an interdisciplinary approach [26].

Studies about functional neuroimaging in specific phobia suggest that the efficacy of psychological treatment in phobic disorders is commonly associated with functional changes in the fronto-limbic brain areas such as the thalamus, amygdala, insula, anterior cingulate cortex, visual cortex, and prefrontal cortex [27,28,29]. This aspect is related to the physiological-emotional factors involved in the maintenance of specific phobias and highlights the dual-path processing model [30]. This model implies the existence of a short pathway for emotional processing of stimuli that involves the direct connection between the thalamus and the amygdala and a long pathway that involves the direct connection between the thalamus, the involved sensory cortex, and the amygdala. In the long processing route, the connection with the prefrontal cortex allows the cortex to provide non-contingent information to the emotion of fear created in the amygdala. This enables the subsequent regulation of a voluntary and planned response to the feared situation or object by the prefrontal cortex on the amygdala [30,31,32]. The regulation process of limbic structures by the prefrontal cortex is also known as the up-down control mechanism [33,34]. A recent meta-analysis on studies based on analysis of the region of interest to appreciate differential activations between healthy subjects and people with specific phobia in various regions of the limbic circuit showed a greater convergence of activations in the right amygdala, insula, and cingulate cortex of phobic patients compared with the controls [35].

Functional neuroimaging techniques such as nuclear magnetic resonance imaging (NMRI) have supported the understanding of how psychological treatments manage to modify neural circuits. They have shown that, in addition to the clinical efficacy of CBT, there is a close relationship between the clinical improvement resulting from therapy and certain brain changes [36,37,38]. There is scientific evidence that exposure to virtual stimuli leads to similar results to those achieved with cognitive-behavioral techniques [39,40]. However, the VRET has shown significant advantages in the longer duration of positive results [39,41] and greater clinical efficacy [21,41].

Previous studies have shown that the pattern of brain activity of people with specific phobias of small animals exposed to real images differs from that of people without this disorder [36], and these differences are maintained after having been treated with CBT + exposure to real images [42]. Another study on the pattern of brain activity resulting from exposing untreated people with specific phobia to phobic stimuli presented in virtual images observed a significant activation of fear processing circuits, as occurs in people exposed to real images [43].

Studies based on self-report tests suggest that the usefulness of VRET is found in the involvement of the frontal structures of the brain to promote self-efficacy and self-instructional capacity, which are important cognitive elements for the best approach to phobic pathology. However, it is necessary to objectively corroborate these therapeutic mechanisms through the information provided by fMRI.

The aim of this study is to compare brain activation from exposure to real images vs. VRET in a CBT treatment program in people with specific phobias of small animals (specifically spiders, cockroaches, or lizards) and whether the results of the treatment program were maintained at 3-month follow-up.

## 2. Materials and Methods

### 2.1. Participants

Computer-based simple randomization was performed, and participants were assigned a correlative numerical code based on the order in which they were contacted. Neither the researchers who recruited and interviewed the participants nor the therapists knew the participant assignment until the time of the first fMRI and treatment sessions, respectively.

Participants were recruited through a public call from the University of La Laguna (Tenerife, Spain) between 2016 and 2018, and all participants provided written informed consent. Participants of the intervention groups were administered the evaluation instruments (see Section 2.2. Instruments) and underwent the first fMRI session with exposure to real or virtual 3D moving images, depending on their group.

A total of 131 adults were randomized, of whom 78 did not receive the assigned intervention for various reasons (not responding, health condition incompatible with fMRI, non-attendance of the appointment for the interview, or fMRI). Finally, 23 received intervention with CBT + VRET (intervention group), and 30 received CBT + real images (active control group); however, there were fourteen dropouts (7 in the intervention group and 7 in the active control group) due to non-completion of therapy due to self-perceived improvement in phobic symptoms or non-assistance post-fMRI. After a realignment analysis of fMRI images was performed, two participants from the active control group and five from the intervention group were excluded because they did not meet the minimum quality criteria for the evaluation and comparison of their neuroimages. As a result, the total sample consisted of 31 participants: 17 in the active control group (real images) and 14 in the intervention group (virtual images) (see Figure 1).

The inclusion criteria for participants were: (1) being an adult with a specific phobia of spiders, cockroaches, or lizards (small animals for which both videos of real and virtual images were developed for exposure purposes); (2) the specific phobia had to be a primary psychological disorder and not be explained by another health condition or phobia; (3) participants who had not received any pharmacological and/or psychological treatment for their specific phobia in the last 12 months; (4) being right-handed; (5) having normal vision; and (6) not having any impediment for a magnetic resonance imaging session (e.g., a metal implant not compatible with fMRI, possible pregnancy, or any other medical condition in which the use of MRI is discouraged).

The most frequent phobic animal was the cockroach and most of the participants indicated that the onset of their phobia was during childhood. Women were the majority, and the mean age of all participants was 33.55 years (*SD* 11.77); there was no significant difference between the two age groups. None of the participants had received therapy prior to this study (Table 1).

### 2.2. Instruments

#### 2.2.1. Clinical Assessments

The S–R (Situation–Response) Inventory of Anxiousness [44] was administered to obtain a participant phobia rating at baseline. A Spanish translation was used. This instrument has a high internal consistency (Cronbach’s alpha = 0.95) and is a 14-item inventory, 5-point Likert scale, that assesses the physiological, cognitive, and behavioral symptoms associated with the response to an anxiogenic stimulus; in this study, the researcher pointed to the phobic animal targeted. This instrument was administered at baseline, post-treatment, and at follow-up (3 months).

To confirm the diagnosis of phobia, participants were asked to answer questions of the structured Composite International Diagnostic Interview (CIDI), Version 2.1, related to specific phobia, social phobia, agoraphobia, and panic attacks [45]. The Latin American and Spanish cultural adaptation was used [46]. They also performed a semi-structured interview that included questions on each specific criterion. Participants diagnosed with specific small animal phobia (F40.218, according Diagnostic and Statistical Manual of Mental Disorders classification code [1]) were included in the study [45].

The Hamilton Anxiety Rating Scale (HAM-A) [47] was administered as a complementary diagnostic test to explore participants’ self-perception of anxiety symptoms experienced in the presence of the phobic animal. The Spanish validation was used [48]. HAM-A showed good interjudge reliability as the intraclass correlation coefficients range from 0.74 to 0.96 [49], and a score of 14 or higher was required to consider participants phobic. This instrument was administered at baseline, post-treatment, and follow-up (3 months).

The Hospital Anxiety and Depression scale (HAD) [50]—specifically the anxiety subscale—was administered according the Spanish validation [51]. It is a short test that allows the participant to obtain a score for depression and other for anxiety, referring to the last week. It was used to obtain a second measure of anxiety in order to confirm that the sample selection was adequate (Cronbach’s alpha = 0.81). This instrument was administered at baseline and post-treatment.

Hand preference was assessed with the Edinburgh Handedness Inventory [52] to determine that all participants were right-handed, with the aim of controlling for the effect of manual dominance on brain activation. This inventory is widely used in research and evaluates manual preference through 10 activities. The cut-off points to determine whether the participants are left-handed, right-handed, or ambidextrous have been established based on statistical criteria [53]. This instrument was administered before the first fMRI.

At the end of the treatment sessions, participants completed the Spanish validation [54] of the Revised Helping Alliance Questionnaire, Patient Version (HAQ-II-PV) [55], to assess the therapeutic alliance. HAQ-II-PV had good psychometric properties and consists of 17-item inventory, 6-point Likert scale (Cronbach’s alpha = 0.88).

All clinical assessments were conducted by the researcher group, and the administration of S-R inventory and HAD scale in the follow-up to obtain a measure of the therapeutic efficacy of the treatment program was carried out by email. This online administration made it impossible to evaluate the HAM-A scale at follow-up.

#### 2.2.2. fMRI Data Acquisition

The nuclear magnetic resonance device used was a GE 3.0 T Signa Excite HD (General Electric, Madrid, Spain GE Healthcare), and the functional images were recorded with Gradient Echo [56] (TR = 2000 ms, TE = 30 ms, FA = 75°, FOV = 25.6, image dimension = 64 × 64 × 32, voxel dimension = 4 mm × 4 mm × 4 mm). The structural images were recorded with T1 FSPGR 3D [56] (TR = 8852 ms, TE = 1756 ms, FA = 10°, FOV = 25.6, image dimension = 256 × 256 × 172, voxel dimension = 1 mm × 1 mm × 1 mm, thickness = 1 mm, space = 0 mm) using the ASSET method (calculated with 35 50° angled images, TE = 2.1 ms, TR = 150 ms, 32 × 32 matrix and 6 mm slice width).

A block design was chosen to present the stimuli in the MRI device, and fMRI sessions were about 11 min per participant. Participants wore glasses that allowed the presentation of images in stereoscopy with paramagnetic isolation (VisualStim digital MRI compatible 3D glasses and graphics card: GeForce 8600GT). Exposure to real moving images of spiders, cockroaches, and lizards was conducted through 3D videos of these animals filmed/recreated virtually in motion. All the images had an identical white background.

### 2.3. Design and Statistical Analysis

The overall design of this study consisted of: (1) an ANOVA to detect differences between groups in clinical instruments scores administered before and after treatment and at three-month follow-up; and (2) analysis of activations recorded in fMRI sessions before and after treatment in both groups using Statistical Parametric Mapping software (SPM 12). A 2 × 3 factor design for the analysis of the results was used (factors: Image and Treatment; levels: real vs. virtual and pre-treatment vs. post-treatment vs. follow-up).

The hemodynamic changes associated with the functional brain activity, that is, the Blood Oxygenation Level Dependent (BOLD) effect [57], was used to study the neuroimaging measures obtained through the fMRI sessions according to the type of exposure (real or virtual images). Both functional and anatomical images were manually reoriented to the anterior–posterior commissure plane before pre-processing. A significance level of *p* < 0.001 was used without any correction algorithm and the criterion to the use of accepting 10 voxels to consider an activation [58]. This criterion is usually used in fMRI studies to eliminate activations that may be false positives for functional voxel dimensions of 2 mm × 2 mm × 2 mm (8 mm^3^), so multiplying the resulting volume of activation by 10 would make it 80 mm^3^. However, in our case, we used voxels of 4 mm × 4 mm × 4 mm (64 mm^3^) and at least 3 contiguous voxels per cluster to consider an activation (i.e., *k-*space data ≥ 3), since multiplying 64 by 3 m the resulting extension volume would be 192 mm^3^, higher than the criteria of frequent use when they use *uncorrected p*. In this way, we assume that the possibility of choosing false positives is controlled.

### 2.4. Procedure

#### CBT Program

Fourteen clinical psychologists were trained in the administration of the psychoeducational program and tests. To guarantee the quality of the sessions and the training of the therapists, biweekly clinical seminars were held with psychologists who were experts in the field to supervise and guide the individual clinical sessions. Each therapist was randomized and blinded to a maximum of five participants.

The CBT program that participants received consisted of a one-hour session per week for 8 weeks, distributed as follows: first session: presentation of the therapist and the program and start of psychoeducation on phobias; second session: presentation and use of the activation of Subjective Units of Anxiety (SUAs) and physiological deactivation with breathing control techniques; third session: practice of breathing control, use of SUAs and presentation of cognitive restructuring; fourth to eighth sessions: review and practice of the contents dealt with in previous sessions and exposure to phobic stimuli with real or virtual images through 3D glasses, with anxiety control for the management of cognitive distortions. The exposure time for the sessions was approximately 27–32 min.

## 3. Results

### 3.1. Clinical Assessments

Table 2 shows the results of intervention and active control groups in the instruments administered. No significant differences were observed between both groups in any instruments at any time evaluated, and there were no differences in the HAQ-II-PV questionnaire regarding the possible effect of the therapist on the outcomes. However, the phobic anxiety scores of both groups had significantly decreased after treatment and remained in the follow-up (S-R *f*_(1,29)_ = 119.029, *p* < 0.001, r = 0.897, d = 4.052; HADanx *f*_(1,26)_ = 5.891, *p* = 0.022, r = 0.430, d = 0.952; HAM-A *f*_(1,29)_ = 75.22, *p* < 0.001, r = 0.850, d = 3.221).

### 3.2. Functional Brain Activation

Regarding the main effect of the treatment, regardless of the type of image presented, the following activations were observed after treatment in both groups: (1) bilateral activation of the thalamus and the lower frontal lobe; (2) unilateral activation of RH in the supplementary motor area, the prefrontal lobe, the precentral gyrus, and the precuneus; and (3) unilateral activation of LH in the insula, cerebellum, and supramarginal gyrus (Figure 2).

Significant differences were observed in the pattern of brain activity after treatment in both groups. The main activation effect caused specifically by exposure to images of the phobic stimulus, regardless of the exposure condition (real or virtual), was bilateral—although predominant in the left hemisphere (LH)—and occurred in the mid-occipital cortex, cerebellum, and hippocampus. Bilateral activation was also observed, but with predominant extension in the right hemisphere (RH) in the fusiform gyrus. Other structures such as the temporal cortex, the calcarine groove, and the lingual gyrus were only activated in the RH. The inferior parietal cortex was the only structure that showed activation exclusively in the LH (Figure 3).

We used an inclusive mask based on anatomical atlas to analyze the indicated regions in targeted structures related to specific phobia. All the structures discussed below showed activity before and after treatment (see Appendix A), so differential activation at each time point will be discussed (i.e., activation before treatment > activation post-treatment and activation post-treatment > activation before treatment). Table 3 show these differential functional brain activations in both groups.

#### 3.2.1. Thalamus

Brain activity detected before treatment in thalamus was significantly higher in the active control group. After treatment, this activity remained bilateral in the active control group and was only recorded in the right thalamus in the intervention group.

#### 3.2.2. Amygdala

After treatment, the differential functional brain activation in the amygdala disappeared in the active control group (*p* < 0.001). However, a significant activation in the left amygdala was observed in the active control group at post-treatment with an extension of *k* ≥ 3 voxels (see Appendix A).

#### 3.2.3. Occipital Cortex

Because the presentation of phobic stimuli involved the animal moving, the visual cortex (Brodmann areas (BA) 17, 18, and 19) was expected to be activated. In the inferior and medial occipital cortex there were no statistically significant differences in brain activity in BA17 and BA18 between baseline and post-treatment in any of the groups. However, differential activity in BA19 was greater at baseline than post-treatment in both groups.

#### 3.2.4. Frontal and Prefrontal Cortex

As regards most anterior areas, a significant bilateral pre-treatment activity that disappeared after treatment was observed in both groups in the orbital frontal cortex. The bilateral activity recorded in the dorsolateral prefrontal cortex in the active control group and in the right dorsolateral prefrontal cortex in the intervention group was significantly higher at pre-treatment than at post-treatment. Cerebral activity recorded in the right ventromedial prefrontal cortex in both groups disappeared after treatment.

#### 3.2.5. Other Brain Structures Involved in Emotional Regulation and Specific Phobias

A change in the location of the post-treatment activity of the virtual image group was observed in the anterior cingulate cortex from a more medial to a rostral area. In the insula and fusiform gyrus, a significantly higher bilateral activity was observed at pre-treatment in both groups. After treatment, a low level of activity was observed in the right insula only in the intervention group.

Finally, baseline and post-treatment activity was observed in the precuneus in both groups, although in different areas. The precuneus was the only structure that showed higher post-treatment activity compared to baseline in both groups. Before treatment, significant activity was observed in the right precuneus in the intervention group and in the left precuneus in the active control group, coinciding with BA7. At post-treatment, a change in laterality reflected in the left precuneus was observed in the activity of the intervention group. In the active control group, post-treatment brain activity was observed bilaterally, coinciding with BA31 and BA23.

## 4. Discussion

The main aim of this study was to determine whether CBT + VRET is of comparable efficacy to CBT + exposure to real images in the treatment of specific phobia to small animals (specifically spiders, cockroaches, or lizards) and whether the results of the therapy are maintained in both treatment modalities during a three-month follow-up period according to the evaluation instruments administered.

Our results join those systematic reviews that indicate the use of virtual reality as a useful tool for the treatment of different mental disorders [59,60,61]. The main advantage of using virtual reality lies in its ability to replicate real stimuli and situations and individualizing the intervention [60,62]. In addition, the VRET can even be considered as a preparation technique for in vivo exposure of the feared stimulus. Although it may seem an expensive technique, some studies show that it is a cost-effective technique for the management of some mental pathologies [63].

The changes observed in the self-report scales assessing specific anxiety associated with small animals suggest comparable therapeutic efficacy between CBT + VRET and CBT + exposure to real images. These results are in accordance with results observed in other studies [22,64].

Regarding the therapeutic alliance (HAQ-II-PV questionnaire), there is literature that shows that the therapeutic effect can explain a high percentage of variance of the treatment effect [65,66,67]. However, in this study, if this effect occurred, it was not significant in any group. Therefore, the therapeutic alliance did not seem to explain the results of the CBT + VRET.

We observed differences between the patterns of brain activity of the both groups at pre-and post-treatment, presumably due to the effect of therapy. In many structures, greater pre-treatment compared to post-treatment activation was observed in both groups. However, this activity was greater in the active control group, perhaps due to the characteristics of the image, since the phobic stimuli used in the active control group were more similar to the phobic stimulus generated by the conditioning, that is, a real animal.

The decrease in the activity of the thalamus after therapy observed is consistent with the results obtained in other studies that found reduced activity in the limbic and paralimbic areas as a result of CBT [68]. However, due to the temporal resolution of the fMRI, in this study, it was not possible to know exactly which thalamic nucleus is activated at the time of exposure to the phobic stimulus. These data support the importance of the connectivity of the limbic and paralimbic circuits in the emotional deregulation of pathological fear [69,70]. The activity observed in the posterior fusiform gyrus is similar to the results of other studies that reported the central role played by this structure in the viso-attentional network related to the awareness of the visual stimulus [71]. In fact, this structure is specialized in high-level vision, for example, in object recognition [72]. The decrease in activation of fusiform gyrus found after therapy in both intervention groups could be interpreted as a lower expenditure of hemodynamic resources by participants when visually attending to the phobic stimulus after therapy [71,72].

The results obtained in the intervention group showed that the activations recorded before and after the therapy changed their location within the cingulate cortex. Medial activity was observed in this structure at pre-treatment, but once the therapy was completed, the activation of the cingulate cortex was recorded in the anterior area. These results seem to point to the research conducted on the specificity of the anterior cingulate cortex depending on its activation. In fact, the anterior cingulate cortex has been linked to a variety of functions, from the processing of rewards to the execution of self-control actions [73]. Some studies suggest that the demands of cognitive control executed by that brain region motivate new learning [74], thus biasing behavioral decision-making towards tasks and strategies that are cognitively efficient in terms of self-care [74,75].

As regards the activity recorded in the precuneus, greater bilateral activation was observed in the participants of both groups after completing the CBT program, as has been observed in previous studies [76]. This structure has been related to episodic memory, the integration of attentional and perceptual processes, visuospatial processing, self-awareness, and the response to emotional stimuli [77,78]. In this regard, the precuneus may act as an emotional regulator that reorganizes the processing of phobic stimuli. It is logical to expect people who have overcome their phobia to have increased their levels of self-efficacy regarding their problem. This is likely to be reflected in an increase in the activity of this structure, since egocentric representations are achieved through a specific network that includes the right precuneus and the angular gyrus. In the results of this study, since the change found in the amygdala before and after therapy was minimal, participants may have self-referenced differently when faced with the phobic stimulus, regardless of the characteristics of the image (real or virtual), which could hypothetically explain the greater energy consumption of this brain structure.

From a theoretical perspective, given the activations found in both groups, the results of this study seem to be consistent with the processing of the long pathway present in the dual pathway model of processing [30]. From a therapeutic perspective, the results coincide with the inhibitory learning model [79,80] more than with the emotional processing model of fear [81,82], because even though a significant decrease in anxiety levels was observed, in the brain, the person would continue to feel afraid. These data suggest that a new adaptive response occurs alongside the initial fear response, rather than replacing it, and the person has learned another way to respond to fear according to the inhibition of the fear response.

Some limitations of this study should be mentioned. Regarding the role of neuroscience in addressing specific phobias, we found some effect on the amygdala after treatment. However, it would be interesting to have performed an MRI test in the follow-up period to determine whether this activity was maintained after therapy and continued to differ from pre-treatment data. This type of study should also be extended to other phobias to determine whether the brain activity of individuals treated for such phobias is similar. Although we performed a semi-structured interview to exclude participants with health conditions explaining their specific phobia, there was not a specific analysis to detect the existence of comorbid disorders.

Further research on functional connectivity is necessary to explore whether the activity of the amygdala upon visualization of phobic stimuli is due to activity in the long or short pathway. In this study, given the technical characteristics of the MRI machine, changes could be inferred from the processing of the long pathway if the activity were observed in the occipital cortex involved in this pathway, but it was not possible to determine what specific input caused the activity of the amygdala. Follow-up fMRI is also especially interesting to explore whether the configuration of the precuneus activations obtained in the present study is permanent or this effect disappears over time. Longitudinal studies with larger samples are needed to explore whether the brain activation pattern of people treated with CBT + VRET and the CBT + exposure to real 3D moving images over time is closer to the brain activation of non-phobic people. Previous comparisons made with people without phobia exposed to the same real images showed significant differences between the patterns of brain activity between participants with and without phobia [42,83].

## 5. Conclusions

CBT + VRET with 3D moving images was comparable in efficacy to CBT + exposure to real 3D moving images, and the results were maintained for up to three months from the end of therapy (according to the data from the instruments administered). The therapeutic benefits modified the brain activity pattern of people treated with a full CBT + VRET program, showing a decrease in the post-treatment activity of structures related to the visual-attentional processing of phobic stimuli. Moreover, the post-treatment activity identified in the precuneus suggests that people with phobia change their way of self-referencing with the phobic stimulus in terms of perception of reality.

The main effect of the treatment was related to activity in more anterior brain areas and related to emotional regulation (e.g., the prefrontal cortex, the insula, and the precuneus). However, the hemodynamic activity of this brain network linked to the emotional fear response was still active, although its response pattern and intensity were modified (e.g., amygdala). This suggests that the therapeutic effect of CBT + exposure to images (real or virtual) could be that the patient has learned another way of responding to fear, inhibiting their fear response.

## Figures and Tables

**Figure 1 jcm-10-03505-f001:**
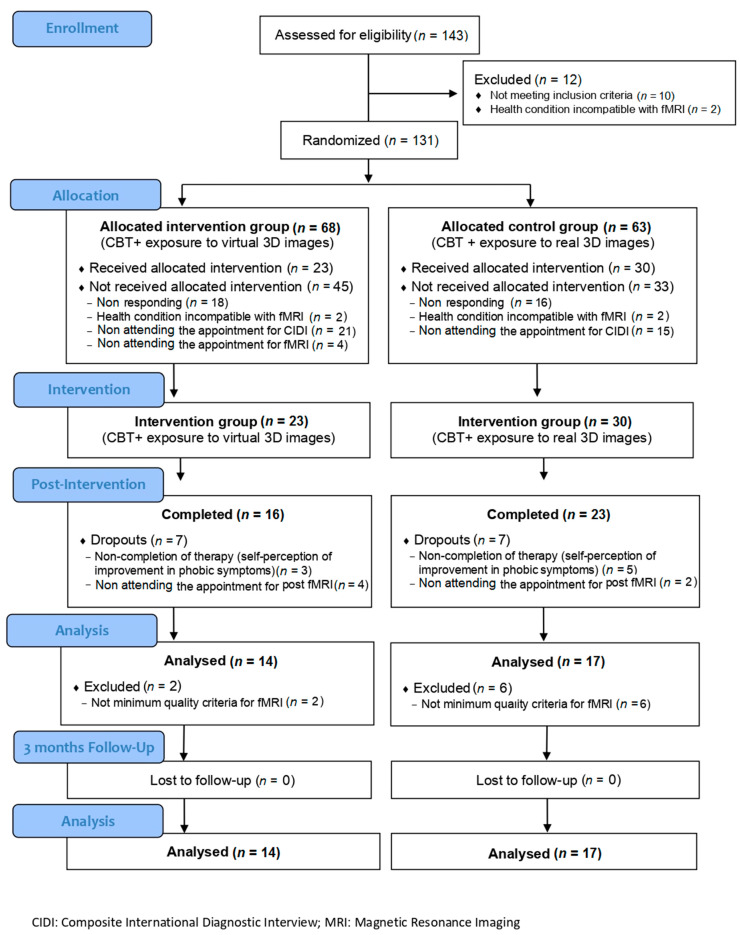
Flow diagram of study enrollment.

**Figure 2 jcm-10-03505-f002:**
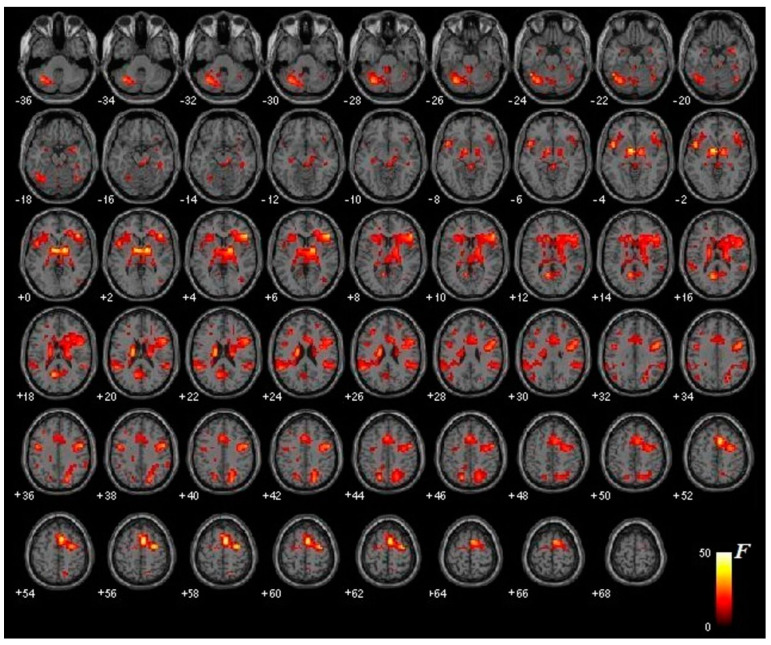
Main treatment effect (*p* < 0.001). The figure shows the differential brain activation of all participants (intervention group and active control group) after treatment when presenting phobic stimuli (regardless of the type of image: virtual or real, depending on the group). The degree of activation ranges from red (not very intense) to white (very intense).

**Figure 3 jcm-10-03505-f003:**
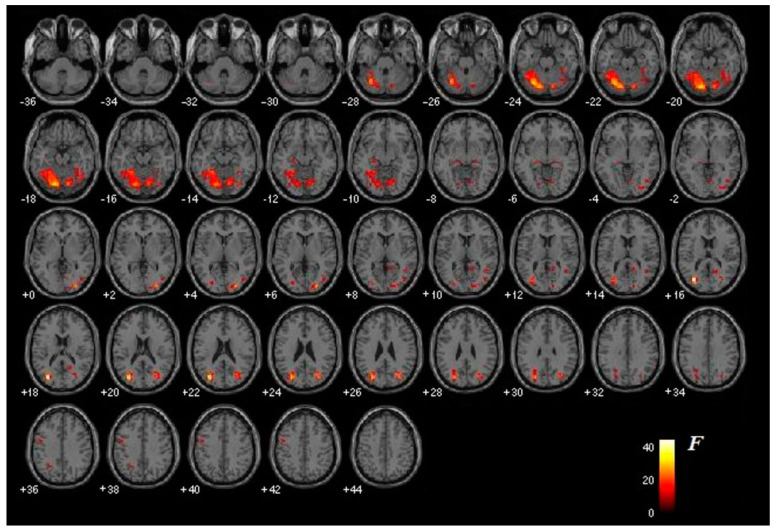
Main image effect (*p* < 0.001). The figure shows the differential brain activation of all participants (intervention group and active control group) caused specifically by exposure to images of the phobic stimulus (regardless of the exposure condition: virtual or real, depending on the group). The degree of activation ranges from red (not very intense) to white (very intense).

**Table 1 jcm-10-03505-t001:** Baseline characteristics of participants.

	Active Control Group	Intervention Group
Participants *(n)*	17	14
Mean age (year)	37.47	29.64
Gender (women)	80%	90%
Phobic animal		
Spiders	24%	21%
Cockroaches	65%	79%
Lizards	11%	0%
Age of onset of phobia		
Childhood (≤11 years)	88%	86%
Adolescence (12–17 years)	12%	14%
Adulthood (≥18 years)	0%	0%

**Table 2 jcm-10-03505-t002:** Clinical assessments.

Instrument	Baseline		Post-Treatment		Follow-up (3 Months)	
	M	sd	M	sd	M	sd
S-R						
Active control group	37.94	6.75	18.6	6.63	21.4	4.8
Intervention group	37.86	5.76	17.21	8.11	20	7.25
HADanx						
Active control group	6.59	4.53	4.75	3.01	4.36	3.39
Intervention group	6.92	2.99	5.93	2.40	6.00	2.39
* HAM-A						
Active control group	17.13	9.13	4.33	2.51	–	–
Intervention group	22.64	6.83	7.36	5.05	–	–
HAQ-II-PV						
Active control group	–	–	73.81	6.23	–	–
Intervention group	–	–	71.36	7.76	–	–

HADanx = Hospital anxiety and Depression Scale, subscale anxiety; HAM-A = Hamilton Anxiety Rating Scale; HAQ-II-PV = Revised Helping Alliance Questionnaire Patient Version; S-R = S-R Inventory of Anxiousness. *** The HAM-A was administered in person at the interview and before the second fMRI session, so this information was not available for follow-up, where the instruments were administered online.

**Table 3 jcm-10-03505-t003:** Differential functional brain activation: before (pre) and after (post) treatment.

	Active Control Group								Intervention Group							
Area	Pre > Post				Post > Pre				Pre > Post				Post > Pre			
	*p*	*t_(58)_*	*k*	*Coordinates* *(x, y, z) ^1^*	*p*	*t_(58)_*	*k*	*Coordinates* *(x, y, z)* ^1^	*p*	*t_(58)_*	*k*	*Coordinates* *(x, y, z)* ^1^	*p*	*t_(58)_*	*k*	*Coordinates* *(x, y, z)* ^1^
Thalamus																
RH	0.000 *	5.03	57	14, −8, −2	–	–	–	–	0.000 *	3.50	7	14, −8, 6	–	–	–	–
LH	0.000 *	4.75	27	−10, −8, 2	–	–	–	–	–	–	–	–	–	–	–	–
Amygdala																
RH	0.003	2.82	3	30, 4, −18	–	–	–	–	0.001 *	3.43	5	30, 0, −18	–	–	–	–
LH	–	–	–	–	–	–	–	–	0.000 *	3.60	5	−26, 0, −22	–	–	–	–
Occipital cortex																
RH	0.003	2.86	3	22, −64, −14	–	–	–	–	–	–	–	–	–	–	–	–
LH	0.000 *	3.52	5	−22, −60, −14	–	–	–	–	0.001 *	3.28	3	−38, −68, −18	–	–	–	–
Frontal orbital cortex																
RH	0.001 *	3.37	5	–	–	–	–	–	0.000 *	3.59	4	38, 24, 2	–	–	–	–
LH	–	–	–	–	–	–	–	–	0.000 *	3.97	4	−42, 20, 2	–	–	–	–
Dorsolateral prefrontal cortex																
RH	0.000 *	3.95	6	50, 4, 34	–	–	–	–	0.000 *	4.00	6	46, 4, 34	–	–	–	–
LH	0.000 *	4.43	6	−46, 4, 30	–	–	–	–	–	–	–	–	–	–	–	–
Ventromedial prefrontal cortex																
RH	0.000 *	4.53	3	18, 8, 50	–	–	–	–	–	–	–	–	–	–	–	–
LH	–	–	–	–	–	–	–	–	0.000 *	3.83	9	−2, 20, 42	–	–	–	–
Anterior cingulate cortex																
RH	0.000 *	3.54	4	2, 20, 26	–	–	–	–	–	–	–	–	–	–	–	–
LH	–	–	–	–	–	–	–	–	0.001 *	3.39	17	−2, 36, 30	0.000	3.54	8	−6, 44, 2
Insula																
RH	0.001 *	3.18	13	38, 24, 6	–	–	–	–	0.000 *	4.56	23	38, 28, 6	–	–	–	–
LH	0.001 *	3.16	8	−30, 16, 2	–	–	–	–	0.000 *	5.49	48	−46, 8, −2	–	–	–	–
Fusiform gyrus																
RH	0.000 *	4.47	7	42, −64, −18	–	–	–	–	0.000 *	3.76	7	38, −36, −14	–	–	–	–
LH	0.000 *	4.50	11	−46, −52, −22	–	–	–	–	0.000 *	3.41	3	−38, −64, −18	–	–	–	–
Precuneus																
RH	–	–	–	–	0.000 *	3.90	6	16, −56, 14	0.000 *	4.21	18	18, −72, 42	–	–	–	–
LH	0.000 *	3.59	3	−10, −60, 46	0.000 *	4.17	9	−10, −60, 18	–	–	–	–	0.000 *	3.90	21	−2, −60, 14

* *p* < 0.001 uncorrected for *k* ≥ 3, ^1^ Coordinates in millimetres of peak voxels in brain regions, LH: Left Hemisphere; RH: Right Hemisphere.

## Data Availability

The data presented in this study are available in supplementary material.

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
