# Peer review of "Changes in Brain Activation through Cognitive-Behavioral Therapy with Exposure to Virtual Reality: A Neuroimaging Study of Specific Phobia"

_jcm, 2021, doi:10.3390/jcm10163505_

Round 1
Reviewer 1 Report
Dear authors, thank you very much for the opportunity to review your work on the very interesting area of the applications of virtual reality on the diagnosis and treatment of certain psychiatric disorders. Phobias are among the most investigated anxiety disorders that VR has an evidence based therapeutic effect. The objective of your research to quantify the brain responses towards real images and VR exposure, of patients following a CBT treatment program, can give further insight regarding the processes of the fearful stimuli.
The major concern is the randomization of your groups. There is a deviation of more than 7 years between the active and intervention groups 37.47 vs. 29.64 mean ages respectively. Nevertheless, the age of onset of the phobias, between the two groups are similar. Since there is evidence that the longer duration of the phobias results to increased alterations of the brain response activities in the areas you studied, could it be in favor of the intervention group and the results might be misleading?
Author Response
First of all, we wish to especially thanks to all reviewers their critical commentaries (and their positive appreciations). We really appreciated their objectives considerations that guided us to a direct enhanced of manuscript. We try to follow all the recommendations and now we think the manuscript has significantly improved.
Please see the attachment.

Reviewer 2 Report
Overview
This paper compares the short-term efficacy as well as changes in brain activity of CBT + VRET and CBT + Exposure to real images in 31 individuals with specific phobia. Authors noted that both groups showed significant decreases in anxiety and phobia scores. Moreover, fMRI findings showed differential cerebral activations at post-treatment, with some regions being associated with increases (e.g., precuneus) and others with decreases (e.g., prefrontal cortex) in activation. This subject is of interest with emerging evidence showing that psychotherapies may modulate neural processes in psychiatric samples. Nevertheless, there are several issues that should be addressed before this paper may be recommended for publication. The comments below are provided to improve the manuscript.
Abstract
Although I understand that the abstract may be limited to word constraints, I nonetheless believe that it may be modified for readers to better grasp the study. In this sense, the background section of the abstract may be reworked to explain the reasoning as to why this study is necessary and clearly state the objectives.
The following sentence “Amygdala remained activated after therapy” should be revised.
Introduction
Globally, the themes discussed in the introduction are pertinent, however, I suggest certain modifications that the authors may desire to bring.
The introduction would benefit from more details and clarity on the prevalence, characteristics as well as available treatments of phobias and more importantly specific phobias. What is the short-term and long-term efficacy (with stated effect sizes) of non-pharmacological interventions to date and the reasoning as to why they may not work (e.g., limitations of current treatments)? More particularly related to CBT, more details on its modalities could be integrated. This would help explain the need for integrating VR to current approaches.
Since the paper is on the efficacy of a VR-based treatment for specific phobia, more information regarding its efficacy in other psychiatric illnesses should be included. This is the case more particularly for anxiety disorders. Although very briefly discussed, the authors should describe evidence of these innovative interventions for the treatment of psychiatric symptoms and describe whether VR is a good modality for treatment. Evidence (with effect sizes) provided from meta-analyses should be integrated.
More details on brain changes following psychotherapeutic interventions, mainly interventions using VR, provided from functional neuroimaging studies should be offered in the introduction. I believe the paragraphs on the functional changes should be refined to clearly define the current state of literature and limitations that are needed to bring forth the necessity of this study.
The objectives of the study should be clarified throughout the manuscript. For instance, the Discussion section states that one of the objectives was to determine whether CBT + VRET and CBT + Exposure to real images is comparable and whether effects were maintained at 3-month follow-up, yet this objective was not stated in the introduction. Moreover, the second objective stated in the Introduction consisting of functional connectivity does not appear to have been feasible as stated in the Discussion section.
Methods
Several elements concerning the sections in Methods require further clarity.
Participants:
Although Figure 1 details the reasonings behind exclusions following treatment allocation, I believe a sentence may be added within the text to detail these cases.
The following exclusion criteria in Figure 1 “Not assistance at fMRI” should be clarified.
I wonder why authors chose to select only individuals with specific phobias to spiders, cockroaches and lizards.
Authors should specify their 4th inclusion criteria. Were both pharmacological and psychological treatments excluded? Moreover, was this criteria for current treatment interventions alone or also within the past 12 months as individuals having followed an intervention within the last year may have had an effect on results?
Although authors note the age of participants in the section, a brief sentence on baseline characteristics should be included and whether there were no significant differences between groups for all baseline characteristics.
I wonder if authors could add details on possible comorbid disorders in their sample.
Clinical assessments
Authors should clearly state in this section whether all the questionnaires were assessed at baseline, post treatment and at follow-up. For better flow, the last paragraph in Procedure (Line 184) may be integrated in this section. Also, details on who conducted the clinical assessments should be included.
The psychometric properties of the questionnaires used in the study should be briefly offered to highlight that the scales have been previously validated in literature.
More details on the HAM-A, HADS and HAQ-II-PV scales should be offered.
Design and statistical analysis
Currently, this section is lacking details on the statistics and models that were employed to measure functional brain activity changes. Moreover, why did the authors choose not to employ any corrections?
Beyond individually analyzing brain changes between pre- and post-treatment, I wonder why authors did not choose to evaluate the statistical associations between changes in clinical outcomes and brain activity changes.
Procedure
This section does not appear to flow within the Methods section and comprises details that may benefit from being integrated within other sections for clarity. For instance, Lines 169-178 should appear in the Participants section, Lines 178-183 in the fMRI section and Lines 184-189 in the Clinical assessments section.
CBT program
This section may benefit from more descriptions on the distinction between both intervention groups. Furthermore, details on exposure modalities and time of exposure should be included.
Results
Clinical assessments
Beyond Mean and SD, Table 2 should include statistical results. Authors may also desire to include the effect sizes of the interventions.
Functional brain activation
Figure 2 and Figure 3 would benefit from a more descriptive title and caption.
Discussion
As stated beforehand, the objectives of the study in the Discussion (Lines 313-317) should be consistent throughout the different sections of the manuscript.
Furthermore, I believe the authors might desire to profound the discussion of their key findings by comparing them more with prior literature. In this sense, more details should be offered on previous literature and whether evidence is comparable. I believe authors should further discuss the utility of VR since both CBT + VRET and CBT + Exposure to real images appear comparable. Since VRET may be more expensive, why should therapists use this modality? Additionally, I believe more information may be offered on fear response that remains with increased activity at post-treatment, and on the potential mechanisms that may explain the efficacy of interventions in this regard.
Authors note several regions that appear significantly different between groups at baseline that should be further discussed in the Discussion section.
I believe the paper would benefit from elaborating on the limitations section and how future studies may remedy these issues. The authors may also add as limitations the small sample size, sample consisting of individuals with solely small animal phobias, lack of correction of statistical analyses, etc.
Author Response

(The authors gave the same response as above.)

Reviewer 3 Report
The topic addressed in the work is very necessary in this area of research. well designed methodologically.
Few previous studies on the subject that make it especially interesting.
It could be included if there is a relationship between the years of suffering from the disorder and the results of the treatment
The objective of the study is highly relevant from a clinical and research point of view. The union of medical examination tests in specific phobias together with a rigorous psychological evaluation is a very interesting study orientation.
As the authors point out (lines 36 and 37), these types of phobias are the most common within community samples, so their study is well oriented. At the same time, they employ a growing, empirically validated, low-cost, and short-term therapeutic orientation, which broadens the clinical utility of the study.
In the introductory section, the previous documentary sources on the VRET are indicated, as well as the empirical bases of its study. Documentary sources have been conveniently included in this section (references 15, 32 and 33)
Participants
The sample has been collected intentionally, probably due to the complexity of the study. In any case, both the procedures used for the selection of the sample and its final composition do not alter the generalizability of the results in relation to previous research and is consistent with the procedure used in previous studies. Perhaps it would be convenient to point out in the conclusions section, the implication that the loss of subjects in the study could suppose due to not having completed the treatment program. In any case, and due to the complexity of the study, the final sample guarantees the relevance of its results.Instruments
A convenient selection of assessment instruments has been made that have been psychometrically verified in previous studies. They are adequate for the objectives of the evaluation, although it would be convenient to indicate the versions in Spanish that have been used.
Design and statistical analysis
The statistical procedures used are correct. Powerful statistics have been used with the usual procedure. The advantages and limitations of the analysis options that have been used in the study are pointed out (lines 163-167).
Procedure
The double-blind procedure used in most studies of this type has been used. The differences in the application of psychological tests in the first and subsequent sessions (online) and the implication of this in the results should be specified.
CBT program
The treatment has been applied in its conventional way
Clinical assessments
It is pointed out that there are no previous differences between the two groups, which is information of great interest in the study.
Discussion
The implications of the study as well as its limitations are adequately discussed.
Author Response

(The authors gave the same response as above.)

Round 2
Reviewer 1 Report
Dear authors thank you for your reply. No further comments.
Author Response
Thank you very much for your reviews and comments. We are really grateful.
Reviewer 2 Report
The authors have adequately addressed my comments and, therefore, I have no further comments. I nonetheless believe authors should revise their paper for English editing, notably the sections that have been added as some sentences may be clarified. I offer the authors some examples:
-Line 16 Authors should reconsider the additional section added. Instead of the term "providing", authors may consider using "addressing", which seems more in line with the sentence.
-Line 56-57 I suggest the following: "causing this procedure to benefit only a limited number of patients and generating considerable dropouts"
-Line 107 Authors should integrate within the sentence the following "...capacity, which are important cognitive elements..."
-Line 126 Authors should remove "instruments described above" since the sentence has been removed from its initial place. Therefore, instruments have not yet been described.
-Figure 1 Instead of "Not attend", authors may consider using the term "Non attending" or "Non-attendance of".
-Line 469 Authors should consider using "Although" instead of "Despite".
Author Response
Many thanks for your comments. We have modified the sentences that you indicated, and we have also made a general revision of the English edition of the manuscript.